# Characterizing Crustose Lichen Communities—DNA Metabarcoding Reveals More than Meets the Eye

**Jacob R. Henrie [1], Brenden M. Thomson [1], Andrew August Yungfleisch [1], Michael Kerr [1] and Steven D. Leavitt [2,\*]**

1 Department of Biology, Brigham Young University, Provo, UT 84602, USA
2 Department of Biology, M. L. Bean Life Science Museum, Brigham Young University, Provo, UT 84602, USA
\* Correspondence: steve_leavitt@byu.edu; Tel.: +1-801-422-4879

**Abstract:** Biodiversity inventories are important for informing land management strategies, conservation efforts, and for biomonitoring studies. For many organismal groups, including lichens, comprehensive, accurate inventories are challenging due to the necessity of taxonomic expertise, limitations in sampling protocols, and the commonplace occurrence of morphologically cryptic species and other undescribed species. Lichen communities in arid regions are often dominated by crustose lichens, which have been particularly difficult to incorporate into biodiversity inventories. Here, we explore the utility of DNA metabarcoding for characterizing the diversity of lichen-forming fungi at a typical crustose lichen-dominated site on the Colorado Plateau in the southwestern USA. We assessed the consistency of independent sampling efforts to comprehensively document lichen diversity, evaluated the capability of minimally trained technicians to effectively sample the lichen communities, and provide a metagenomic-based inventory of lichen diversity, including representative sequence data, for a diverse, crustose-dominate lichen community on the Colorado Plateau. Our results revealed that crustose lichen communities in the southwestern USA are more diverse than traditionally thought, and community metabarcoding is a promising strategy for characterizing the lichen-forming fungal diversity more thoroughly than other methods. However, consistently sampling the diversity of crustose lichen communities, even at small spatial scales, remains difficult. Interpreting these results within a traditional taxonomic context remains challenging without the use of vouchers.

**Keywords:** amplicon sequencing; biodiversity; biomonitoring; ecological sampling; Illumina; internal transcribed spacer region (ITS); inventory; ITS2; semi-arid





## 1. Introduction

Rapid and accurate biodiversity assessments are a major goal in conservation research [1–3]. However, most traditional methods of inventorying biodiversity are slow and costly, often requiring specific training and years of experience [4]. While lichens have been used for decades to effectively monitor air quality and ecological disturbances [5,6], comprehensive lichen inventories are notoriously challenging [7–9]. Accurate lichen identifications often require specific training, a familiarity with regional lichen diversity, and access to microscopy equipment and chemical tests [10,11]. While phylogenetic and genetic analyses may be performed on occasion, identification is assigned chiefly by the presence of phenotypic traits. Charismatic macrolichens, e.g., large, foliose or three dimensional lichens, often epiphytes, are frequently targeted to minimize the challenges with sampling and the specimen identification of crustose microlichens, e.g., those that are intimately associated with the substratum and generally difficult to remove [12,13]. Crustose lichens rank among the slowest growing lichen forms and display a wide range of variation in their attachment to the substratum, with some occurring as endoliths, but most have a defined upper cortex, algal layer, and medulla [13] (Figure 1). Lichen monitoring programs

often rely on sampling only a subset of lichen communities, rather than complete surveys of lichen diversity [14,15]. These simplified sampling approaches may focus on known indicator species [15], targeting "morphospecies", rather than full, traditional identifications [2], targeting a subset of macrolichens, as implemented in the USDA Forest Service Forest Inventory Analysis program [14], and other strategies.

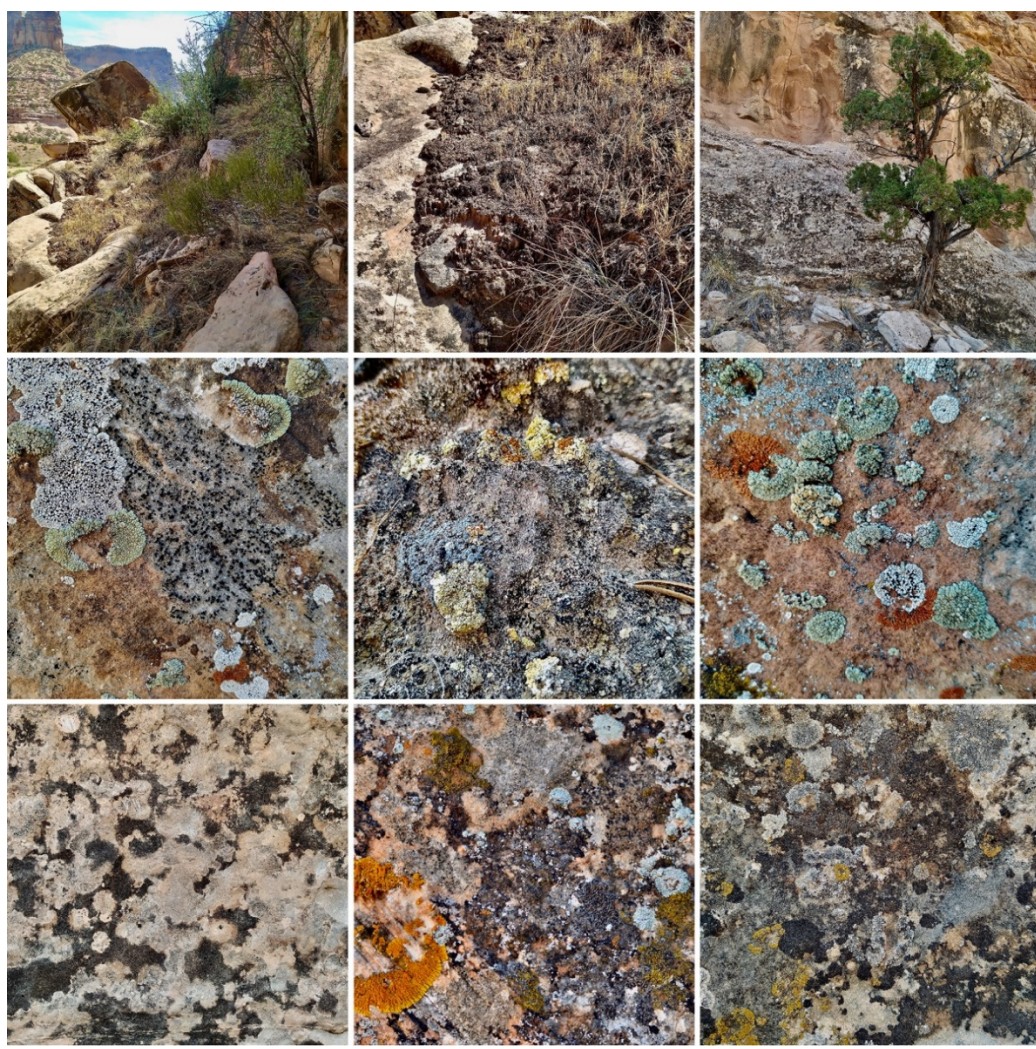

**Figure 1.** Crustose lichen-dominated habitat on the Colorado Plateau, Buckhorn Wash, Utah, USA. The top panel depicts upland slopes covered with large, fallen sandstone boulders, a protected biological soil crust community, and a stable sandstone cliff face, left to right. The lower six panels depict some of the variation in crustose lichen communities occurring within the scale of meters at the Buckhorn Wash site selected for this study.

Incorporating macrolichens into bio-monitoring and conservation research is well established [14]. However, many habitats in the semi-arid southwestern region of the United States support only a limited number of macrolichens [16], but harbor rich crustose/microlichen communities [17]. On the Colorado Plateau in the southwestern USA (Figure 2), regional climatic variation and complex topographic gradients play important roles in shaping lichen communities [16]. Our current understanding of how lichens respond to climate change, land-use practices, and other ecological disturbances is limited by a lack of quantitative, evidence-based derivations [18,19]. Incorporating a more complete perspective of lichen diversity, beyond that of macrolichens, in areas that are dominated by crustose/microlichen communities (e.g., Figure 1) may play an important role in conservation and bio-monitoring research, particularly in sensitive dryland ecosystems [20].

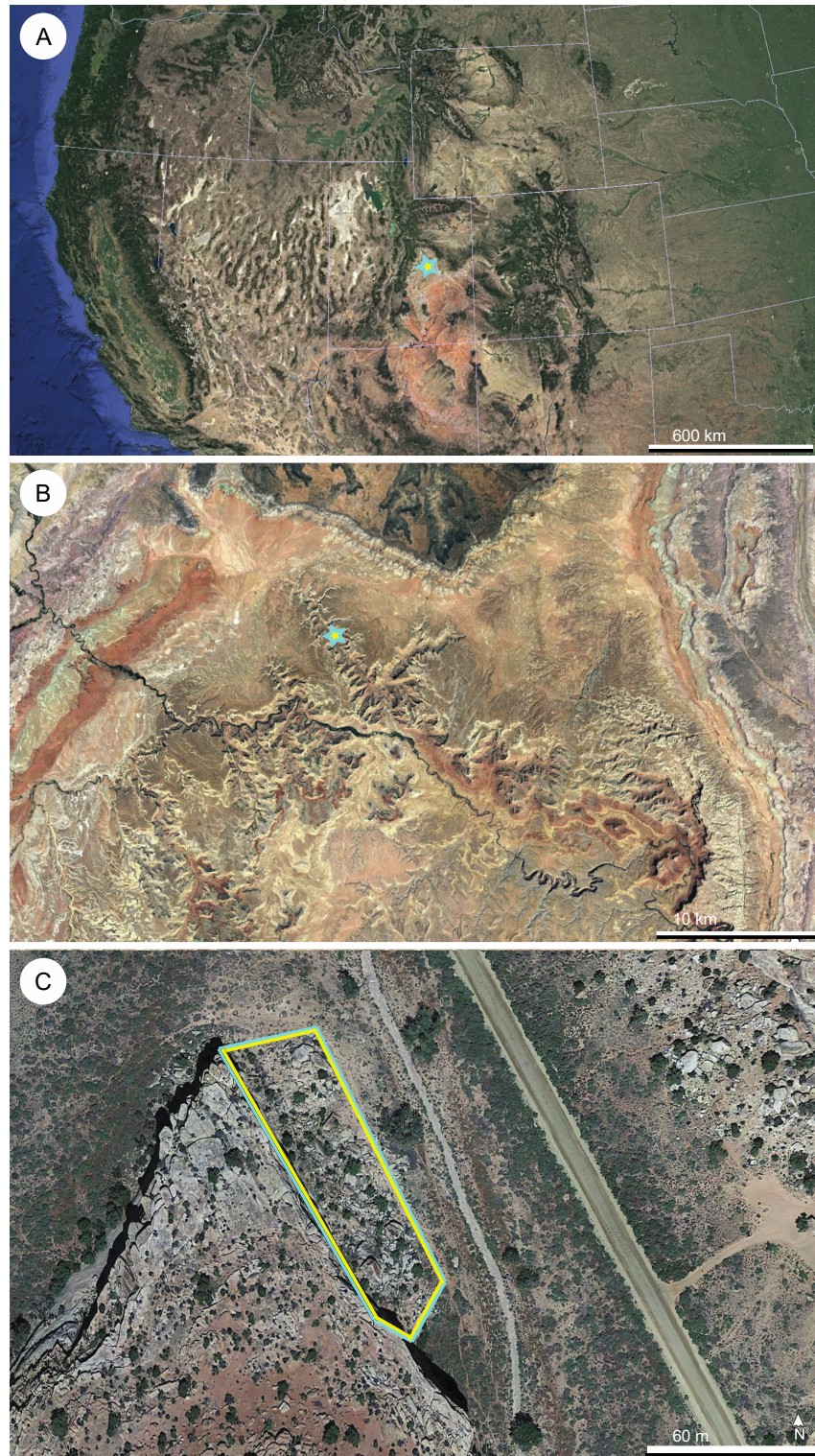

**Figure 2. (previous page).** Overview of sampling location in Buckhorn Wash, Emery County, Utah, USA. (**A**) Blue-outlined star indicates sample site within the broad context of western North America and the location of the site on the northwestern edge of the Colorado Plateau; (**B**) satellite view of the "Semiarid Benchlands and Canyonlands" ecoregion in the San Rafael Swell, with the sample site being marked in a blue-outlined star; (**C**) the location of the sample site in relation to Buckhorn Wash Road, with the targeted area being outlined in blue/yellow.

An alternative methodology for characterizing fungal diversity that has gained recent significance is DNA metabarcoding [21–24]. Metabarcoding approaches for cataloging biodiversity present a potential solution to many of the intrinsic costs and bottlenecks of traditional inventory methods. Metabarcoding relies on using representative samples from which DNA is extracted, amplified, and sequenced, before their taxonomic assignment using a reference database to identify the taxonomic affiliation of the genetic clusters that are recovered in the sample. Studies typically rely on the standard DNA fungal barcode—the internal transcribed spacer region (ITS)—and can target specific subregions, e.g., ITS1 or ITS2, or larger portions of the nuclear ribosomal cistron [21,25]. Current approaches rely on a variety of high-throughput sequencing methods and platforms to characterize the range of genetic variation within the metagenomic samples [22]. Following best practices is essential for ensuring the integrity of the metagenomic samples [21], while also facilitating sample collection and specimen identification by non-specialists, e.g., appropriately trained technicians.

Recent lichen-related metabarcoding studies demonstrate a significant potential for advancing lichenological research [26–30], thus presenting potential solutions for expediting crucial lichen biodiversity inventory research. Importantly, these studies highlight the utility of metabarcoding to characterize lichen diversity more fully and efficiently than can be accomplished using traditional methods [26–30]. However, understanding how well metabarcoding approaches capture the full range of lichen diversity, especially among independent sampling efforts, is not well understood [26]. Therefore, we explore the utility of DNA metabarcoding for characterizing the diversity of lichen-forming fungi at a typical crustose lichen-dominated site on the Colorado Plateau in Utah, USA. Specifically, we aim to (i) assess the consistency of independent sampling efforts to comprehensively document lichen diversity at a small scale, (ii) evaluate the potential capability of minimally trained technicians to effectively sample lichen communities, and (iii) provide a metagenomic-based inventory of lichen diversity, including representative sequence data, for a diverse, crustose-dominate lichen community on the Colorado Plateau.

## 2. Materials and Methods

### 2.1. Site Selection and Field Methods

To explore the efficacy of DNA metabarcoding to compile comprehensive lichen biodiversity inventories in crustose lichen-dominated communities, we chose a location in the San Rafael Swell in central Utah, USA, which represents a typical habitat in the "Semiarid Benchlands and Canyonlands" ecoregion (Figure 2). While the San Rafael Swell is known to support diverse crustose lichen communities, systematic surveys of lichen diversity are generally lacking. In addition, culturally significant archeological sites are present throughout the region, and characterizing lichen diversity may help to protect and contextualize these important sites [31–33]. For this study, we selected a site ca. 2.5 km from the Buckhorn Wash Rock Art Panel [34], on the west side of Buckhorn Draw Road, at the base of the cliffs, among fallen rocks and boulders, and at an altitude of ca. 1985 m ASL (Figure 3). The site is approximately 0.4 hectares and is bounded by the following coordinates: 39.1364, −110.7010 (south boundary) and 39.1373, −110.7015 (north boundary). On the east, the site extends to the ephemeral Buckhorn Wash and on the west, the site is bounded by sheer sandstone cliffs. The rock is predominatly Wingate Sandstone (Eolian, 206–199 Ma).

Common woody vascular plants at the site included *Artemisia nova*, *Atriplex confertifolia*, *Berberis fremontii*, *Cercocarpus intricatus*, *Ephedra viridis*, *Fraxinus anomala*, *Juniperus* spp., *Pinus edulis*, *Populus fremontii*, and *Sarcobatus vermiculatus*.

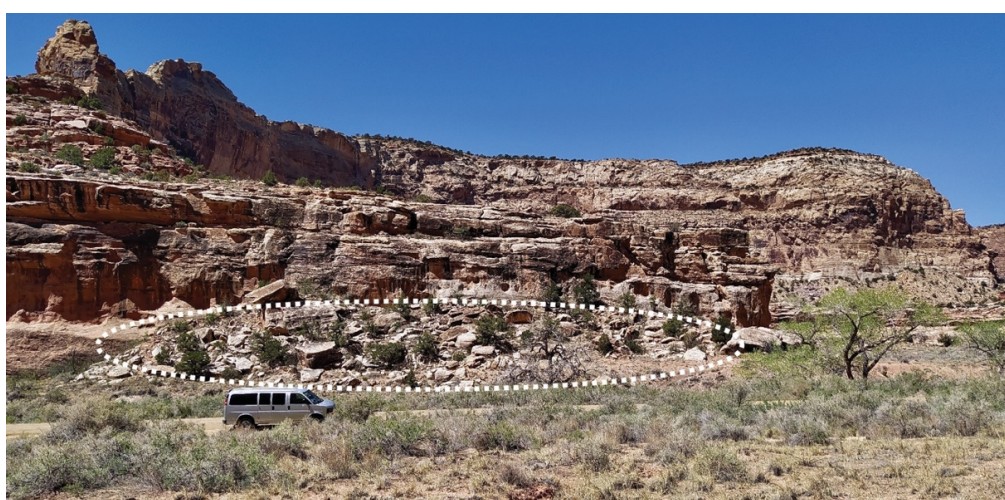

**Figure 3.** The sample site in Buckhorn Wash, Emery County, Utah, USA, with the targeted area outlined by a dashed, white line.

For the field component of this study, the overarching aim was to collect representative bulk samples that were comprehensively representative of the lichen diversity at the site in Buckhorn Wash. Sampling was performed on two separate occasions in 2022. In March, the sampling was carried out by three separate pairs of collectors. Two of the three pairs comprised minimally trained technicians as both members, while the third pair was made up of a minimally trained technician and a professional lichenologist (S.D.L.). Prior to the fieldwork, all participants were briefly trained on collection techniques, common substrates for lichens, and specific protocols for the study. Participants were instructed to intuitively meander the site and collect lichens for two hours or until 15 min had gone by since the last tentative new lichen was observed and collected. Based on the field observations, using a $10\times$ hand lens, small, similarly sized portions of lichen thalli were sampled from all potentially different lichens using sterilized tweezers to pick or scrape material for bulk, metagenomic analyses. The samples were taken from lichens that were on boulders, woody vascular plants, soil, and the surrounding cliff faces. The bulk samples were collected while they were dry and placed directly into a sterile Nasco Whirl-Pak 18 oz. collecting bag (Nasco, Fort Atkinson, WI, USA), returned to the lab within four hours of collecting, and stored at $-80\,^\circ$C until the DNA extraction step was performed. To assess consistency across time and unique collectors, a second round of sampling occurred in May 2022. In this case, the sampling team was comprised of two minimally trained technicians that were not involved in the initial sampling, plus the same professional lichenologist (S.D.L.). A summary of bulk samples is provided in Table 1. No vouchered specimens were collected.

*2.2. Laboratory Methods*

DNA was extracted from each of the four bulk community samples, separately. Community samples (Table 1) were homogenized using sterilized mortar and pestles; DNA was extracted from ca. five to nine g. of homogenized material from each sample using the PowerMax Soil DNA Isolation Kit (Qiagen, Hilden, Germany). In the cases where a high biomass was collected from conspicuous, large lichens, these fragments were subsampled in the lab to qualitatively normalize the approximate size of the lichen fragments before DNA extraction. To characterize the range of lichen-forming fungal diversity in each sample, we amplified a portion of the internal transcribed spacer region—the standard barcoding region for fungi [3]—from each meta-community DNA extraction. Specifically, the hypervariable ITS2 region was amplified using a polymerase chain reaction (PCR) with the primer pair ITS3F (GCATCGATGAAGAACGCAGC) and ITS4R (TCCTCCGCTTATTGATATGC). The PCR products were sequenced at RTL Genomics (Lubbock, TX, USA) using a $2 \times 300$ paired-end sequencing technique on the Illumina MiSeq platform.

**Table 1.** Summary of the four distinct sampling efforts, including the sample name, a description of the participants within each team, and number of reads resulting from the amplicon sequencing using Illumina's MiSeq, and the number of lichen-forming clusters and candidate species represented by ≥5 reads.

| Sample Name | Sampling Team | # of Reads | # Lichen-Forming Fungal Clusters/Species |
|---|---|---|---|
| "TECH1" | technician A & technician B—March 2022 | 48,051 | 119/93 |
| "TECH2" | technician C & technician D—March 2022 | 45,328 | 120/102 |
| "PROF1" | professional (SDL) & technician F—March 2022 | 53,492 | 191/142 |
| "PROF2" | professional (SDL), technician G & technician H—May 2022 | 127,049 | 320/158 |
| Total | composite of all sampling efforts | 273,920 | 473/212 |

*2.3. Data Analyses*

FROGS v3.2 (Find, Rapidly OTUs with Galaxy Solution) was used to analyze the ITS2 amplicon metabarcoding data [35,36]. FROGS v3.2 is a standardized pipeline containing a set of tools that are used to process amplicon reads that have been produced from Illumina sequencing. We followed the protocol that is outlined in [36]. In short, paired-end reads for each sequence in the data were merged, the primers were trimmed, and the unmatched sequences were discarded in the preprocessing step. The merged reads were then filtered using the swarm clustering tool; the clusters were first formed using aggregation distance clustering while it was set to 1, as per the guidelines for v3.2. Chimeric sequences were then removed using the chimera removal tool by implementing the default parameters. The FROGS v3.2 filtering tool was then used to remove the low abundance clusters by setting the minimum proportion of the sequences per OTU to 0.000005. All of the remaining clusters were filtered using the ITSx tool to ensure that the clusters met the requirements for the ITS2 region in preparation for the taxonomic affiliation step. An initial taxonomic assignment of the sequences was completed by comparing the sequences to those in the UNITE 8.3 database [24]. All analyses were performed using the Migale Galaxy Server (https://galaxy.migale.inra.fr, accessed on 21 July 2022).

Subsequently, the clusters that were assigned to lineages comprised of lichen-forming fungi were compared against a custom regional ITS database of lichen-forming fungi for the Intermountain West—BOLD project LIMW. Previous work suggests that family-level multiple sequence alignments (MSAs) provided reasonable MSAs for identifying divergent species-level lineages in other nearby lichen communities [37]. In this study, the family-level MSAs were generated from the genetic clusters from the ITS2 short-read data from Buckhorn Wash and they were aligned with full length ITS sequences from the BOLD project LIMW using the program MAFFT v7 [38]. We implemented the G-INS-i alignment algorithm and the '1PAM/K = 2' scoring matrix, with an offset value of 0.1, and the 'unalignlevel' = 0.4, and the remaining parameters were set to their default values. The family-level ITS MSAs were analyzed under a maximum likelihood (ML) criterion and this was implemented using the IQ-TREE v2 [39], with 1000 ultra-fast bootstrap replicates [40], and the best-fitting substitution model for the entire ITS region was selected using ModelFinder [41]. The trees were visualized using FigTree v1.4.4 [42]. No attempt was made to compile or include closely related ITS sequences from publicly available databases, e.g., NCBI's GenBank, because it is unlikely that all close relatives of a given species would occur in the study area, and restricting the phylogenetic inference to locally occurring sequences may improve the discrimination of distinct species-level lineages [43]. In cases where sequences were not recovered within the monophyletic candidate species

that was represented in the custom regional database from BOLD, BLAST searches against GenBank were performed to identify the most similar sequences and potentially inferred taxonomic identity. However, we note that using sequence similarity to infer taxonomic identity and other issues that are associated with publicly available sequences come with significant caveats [24,44]. Venn diagrams assessing the similarities of the genetic clusters and the candidate species between the different sampling efforts were made using the web-based tool InteractiVenn [45].

## 3. Results

The field sampling efforts resulted in five to nine grams of bulk community lichen material per sample, with each sample representing fragments from hundreds of distinct lichen thalli. The Illumina amplicon sequencing resulted in 45,328 to 127,049 reads per sample (Table 1). All reads are available in the NCBI's Short Read Archive, under PR-JNA866119. The reports of the FROGS pipeline, e.g., preprocessing, chimera removal, OTU filter, and ITSx, are available in Supplementary Files S1–S4. In summary, 10.4% of the sequences, representing 60.6% of the clusters, were excluded as they were chimeric sequences (Supplementary File S2). Of the remaining clusters, 82.5% (3620) were excluded, as they did not meet the minimum proportion threshold, e.g., low-abundance clusters. The remaining 769 clusters comprised 96.3% of the sequences that passed the chimera filter (Supplementary File S3). From these, forty-seven additional clusters were excluded, that did not pass the ITSx filter, resulting in a total of 722 clusters retained for taxonomic assignment (Supplementary File S4).

Across all samples (Table 1), 722 clusters were assigned to 20 classes of Fungi, 53 orders, 95 families, 156 genera, and 252 species using the FROGS affiliation pipeline based on the UNITE 8.3 fungal database (Figure 4; Supplementary File S5). The different sampling efforts variably captured fungal diversity across all taxonomic levels, with the two professional-led sampling efforts—"PROF1" and "PROF2"—capturing the highest levels of fungal diversity (Figure 4). Nearly two-thirds of the 722 clusters were inferred to be derived from lichen-forming fungi (Table 1; Supplementary Files S6). Because this study focuses on lichen-forming fungi, other fungi were not considered further.

A total of 473 clusters were inferred to represent lichen-forming fungi in the FROGS affiliation step. Our subsequent custom species-level taxonomic assignments which were created using the BOLD regional lichen ITS database resulted in a total of 212 candidate species (Supplementary File S6), with the individual sampling efforts recovering between 93 and 158 candidate species each (Table 1). Sequences for all the clusters and the provisional taxonomic identifications are provided in Supplementary File S6. The taxonomic distribution of the clusters and candidate species is reported in Table 2. Acarosporaceae (110 clusters/44 candidate species), Verrucariaceae (88/38), and Lecanoraceae (80/35) were the most diverse families that were sampled at Buckhorn Wash, while both Cladoniaceae and Lichinaceae were represented by only a single cluster. Overall, ca. 63% of the candidate species that were recovered at the Buckhorn Wash sites were present in the BOLD regional lichen ITS database. Most of the novel genetic clusters/candidate species that were represented by a single sample came from the follow-up sampling team—"PROF2".

We note that a cluster representing the fruticose lichen-forming genus *Usnea* was recovered but no physical specimen was observed during collecting by any of the participants.

Of the 473 clusters representing lichen-forming fungi, 164 clusters—ca. 35%—were shared among two or more samples (Figure 5A).

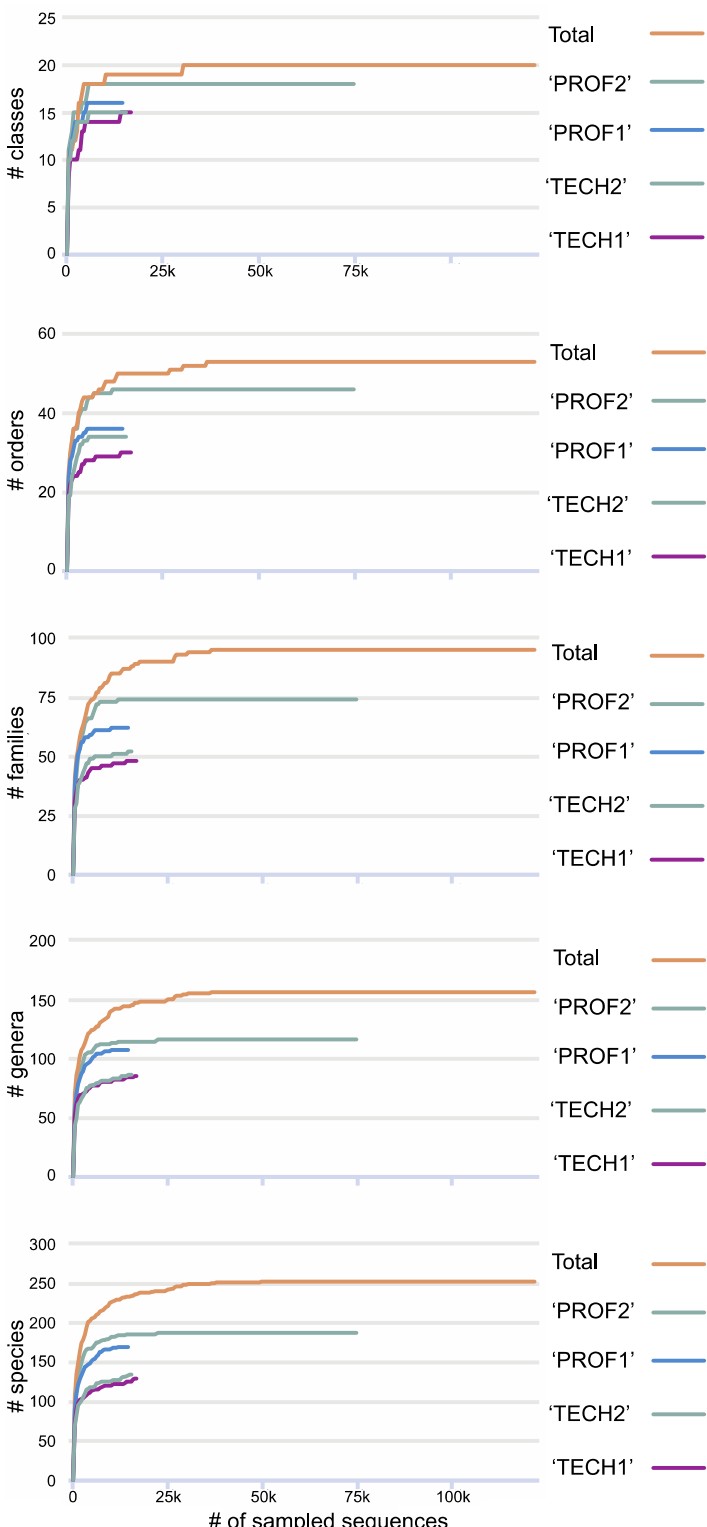

**Figure 4.** Accumulation curves at various taxonomic levels—class, order, family, genus and species—for each of the four sampling efforts (see Table 1 for summary). Accumulation curves for the pooled samples are represented by the orange-colored lines that are marked "Total".

**Table 2.** Taxonomic distribution of genetic clusters and candidate species that were recovered at Buckhorn Wash, Emery County, UT, USA. The proportion of candidate species that was sampled in Buckhorn Wash but was not represented in the BOLD regional DNA reference library (BOLD project LIMW) is reported parenthetically next to the total number of candidate species in each family.

| Family | Clusters | Candidate Species |
| --- | --- | --- |
| *Acarosporaceae* | 110 | 44 (0.52) |
| *Caliciaceae* | 25 | 11 (0.55) |
| *Candelariaceae* | 51 | 9 (0.33) |
| *Cladoniaceae* | 1 | 1 (1.0) |
| *Lecanoraceae* | 80 | 35 (0.31) |
| *Lichinaceae* | 1 | 1 (1.0) |
| *Megasporaceae* | 18 | 12 (0.17) |
| *Parmeliaceae* | 3 | 3 (0) |
| *Physciaceae* | 28 | 18 (0.28) |
| *Placynthiaceae* | 2 | 2 (1.0) |
| *Psoraceae* | 3 | 3 (0.33) |
| *Ramalinaceae* | 10 | 7 (0.43) |
| *Stereocaulaceae* | 2 | 1 (0) |
| *Teloschistaceae* | 40 | 19 (0.10) |
| *Thelotremataceae* | 3 | 2 (0) |
| *Trapeliaceae* | 2 | 2 (1.0) |
| *Verrucariaceae* | 88 | 38 (0.34) |
| Unknown | 6 | 4 (1) |
| Total | 473 | 212 (0.37) |

One hundred thirty-five species—ca. 65% of the total species—were shared among two or more samples, but just over 25% were shared among all samples (Figure 5B). The two samples that included the same professional lichenologist shared 102 species. The candidate species that were observed in only a single sample are highlighted in Supplementary File S6.

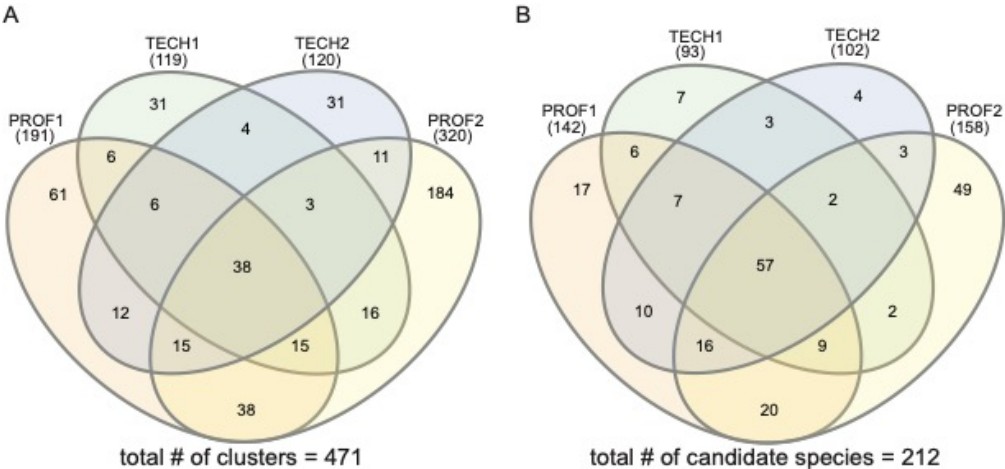

**Figure 5.** Venn diagrams comparing the four separate samples from the Buckhorn Wash site. (**A**) comparisons of clusters among separate samples; (**B**) Comparisons of candidate species among separate samples.

## 4. Discussion

Our metabarcoding-based lichen inventory in the semiarid Southwest, USA revealed an unexpectedly rich and diverse community of lichen-forming fungi across a small spatial scale (Figure 3), with over 473 genetic clusters representing 212 candidate species (Table 2—see Supplementary File S6 for the full list). These results suggest that crustose lichen communities on the Colorado Plateau in southwestern USA are likely even more diverse than traditionally thought [46]. Furthermore, we show that community metabarcoding is a promising strategy to more thoroughly characterize lichen-forming fungal diversity, by capturing not only species-level diversity but also providing unprecedented insight into their genetic variation using the standard DNA barcoding marker [21]. The findings here are consistent with other recent studies that have incorporated DNA sequence data into lichen inventories in the southwestern USA [26,37,47]. While metabarcoding provides a powerful tool for characterizing lichen-forming fungal diversity [27], our studies show that consistently sampling this diversity by employing individual teams remains challenging. Furthermore, the lack of physical vouchers creates significant limitations when interpreting these results within a traditional taxonomic context. Below, we discuss the implications of our findings.

Recently, the best practices for metabarcoding of fungi have been proposed [21]. Coupling these best practices with standardized bioinformatics pipelines, such as FROGS [36], facilitates unprecedented consistency in metabarcoding studies of fungi, including lichen-formers. The DNA metabarcoding strategy that was implemented here provides one possible approach to rapidly provide quantitative lichen diversity data. With modest field efforts (ca. 2 h), we have shown that two pairs of technicians were able to capture nearly 60% of the overall species diversity (126 of 212 total species) with minimal training, while each pair alone captured just less than 50% of the total species diversity (Table 1; Figure 5B). We predict that with a reasonable amount of additional training and experience, these technicians would likely be able to sample a greater proportion of lichen diversity with similar efforts in the field. Our results show that technicians and potentially minimally trained citizen scientists can make significant contributions to lichen biodiversity inventory research, and their efforts should not be discounted [48]. Field collections by citizen scientists/technicians would require a reasonable amount of basic training and minimal supplies, which would allow for a broader range of individuals to contribute to lichenological research without having years of professional experience.

The laboratory and sequencing costs were also relatively modest for the protocol that was implemented in this study—ca. $150 USD/sample. The straightforward field sampling and low costs of this study could also encourage land managers to generate inventories of lichen-forming fungi more consistently to better inform decisions on land management strategies and conservation. However, the interpretation of the results from metabarcoding studies are best contextualized within evolutionary and contemporary lichen taxonomy frameworks. Therefore, capitalizing on expertise in lichen taxonomy and phylogenetics will promote the appropriate interpretation in this type of biodiversity research.

For comparison, the results of the molecular-based metabarcoding study at the Buckhorn Wash site were compared with the herbarium records of the Consortium of North American Lichen Herbarium (CNALH; https://lichenportal.org/cnalh/; accessed on 8 August 2022) of lichens that were collected throughout the entire San Rafael Swell in central Utah, USA (the Buckhorn Wash site occurs within the San Rafael Swell). A polygon-constrained search of herbarium records from the San Rafael Swell region—ca. 4500 km$^2$—produced 309 records documenting 86 taxa, spanning collections from the 1950's to the present (Supplementary File S7). Given the rate of characterizing lichen diversity using traditional voucher-based approaches, our ability to create meaningful, regional baseline data and effective monitoring approaches in the face of contemporary ecological changes is likely limited [20,49]. We encourage the use of alternative strategies, such as the DNA metabarcoding approach that was implemented here, to expedite lichen biodiversity inventories. We predict that our sampling at Buckhorn Wash, with 212 candidate species

(Table 1), represents only a fraction of the overall regional lichen diversity. Implementing a similar DNA barcoding approach at sites throughout the region would quickly and effectively transform our perspective of lichen diversity, thus providing crucial baselines for conservation and monitoring change [8].

By nature, the bulk sampling of lichen communities facilitates the collection of multiple individuals of the same species, especially in composite samples representing different collection teams or individual collectors. Therefore, intraspecific genetic diversity can be approximated within a site, with the genetic clusters serving as a kind of "genetic voucher". In this study, we found that individual teams collected largely non-overlapping genetic clusters even though they were collecting the samples in close proximity (Figures 1 and 5A), demonstrating that high levels of genetic diversity occur even at a local scale in Buckhorn Wash. If these clusters do, in fact, represent high levels of intraspecific diversity, it suggests that there have been multiple colonization events and the long-term persistence of closely related conspecifics/congeners (Table 1). This has crucial implications for exploring questions relating to lichen community assembly that would not be possible with a traditional site inventory that reports only species composition [50–52].

However, we caution that DNA metabarcoding should not completely replace traditional, voucher specimen-based inventories. Rather, both methods should be used to complement one another to improve our understanding of biodiversity and provide physical records of lichen diversity [26,53]. Metabarcoding studies typically rely wholly on reference barcoding libraries for comparison. However, all currently available databases, including the UNITE database [24], are incomplete and lack the information that is necessary to successfully identify all of the genetic clusters [23]. For example, the conspicuous lichen *Squamarina lentigera*—a common terricolous lichen in the Colorado Plateau region—was observed at the Buckhorn Wash site, but its presence was not inferred from the metagenomic data. ITS sequences representing *S. lentigera* are presently not available in GenBank nor in our regional database, and this lichen may be represented by one of the 'unknown' clusters (Supplementary File S6). While our efforts to generate a regional DNA reference library for the Intermountain West, USA—BOLD project LIMW—allowed us to link most of the genetic clusters from the candidate species in Buckhorn Wash with vouchered specimens from previous work, 37% of all candidate species from Buckhorn Wash were not represented in our custom regional ITS reference library.

In addition to incomplete reference libraries, analyses of short read data and the interpretation of the results can be subjective [23]. We highlight that metagenomic data can be integrated and reused in subsequent studies, making these data a powerful resource for long-term monitoring and conservation research [54–56]. Similarly, molecular-based taxonomic assignments can be easily revised with updated and improved DNA reference libraries [24], thus facilitating legitimate cross-study comparisons [56]. In the example of the expected lichen *Squamarina lentigera*, which was not inferred from our metagenomic data (mentioned in the previous paragraph), the unidentified "Cluster 79" is a reasonable candidate for the species as it was collected by each sampling group and found in relatively high-read abundance. With the subsequent inclusion of the reference sequences from this taxon, its presence in the metagenomic data can easily be investigated.

Our bulk sampling approach, and other eDNA approaches, may also amplify the DNA from lichen-forming fungi that are not established at a site, such as the DNA from propagules, spores, and other sources [26]. In fact, in this study, we observed short reads that were inferred to represent the fruticose lichen genus *Usnea*, a lichen that had not been observed during field work and was not expected to occur in this habitat. Determining which genetic clusters from metabarcoding samples represent established lichens vs. propagules/spores requires careful validation with field observations or representative vouchers. Since no voucher specimens were collected as part of this study, unknown genetic clusters or potentially new species cannot be investigated beyond the molecular-based comparisons.



Incorporating voucher specimens into lichen inventories can help to mitigate the potential weaknesses of surveys that rely exclusively on DNA metabarcoding [37,47], specifically the problems with limited databases and previously unknown species. Given the high level of species diversity that is observed in this study—212 candidate species—comprehensively collecting vouchers may be challenging in practice. Prioritizing the collection of vouchers that represent the most poorly known groups of lichens during the bulk sampling step may help to provide representative vouchers. Alternatively, subsequent targeted sampling based on the results of an initial metagenomic study may help to direct the collection of crucial vouchers. In other regional lichen inventories, vouchered specimens have been collected, and from these, samples were selected for DNA sequencing [26,37,47]. Ultimately, researchers must carefully consider the aims and purpose(s) of their inventory work to try balance the practical constraints with the aims of their research [57,58].

**Supplementary Materials:** The following supporting information can be downloaded at: https://www.mdpi.com/article/10.3390/d14090766/s1, Supplementary File S1: the FROGS v3.2 preprocessing report, including a summary of filtered reads and details on merged sequences; Supplementary File S2: the FROGS v3.2 chimera removal report, including the proportion of clusters and sequences that were removed, in addition to chimera detection by sample; Supplementary File S3: the FROGS v3.2 OTU filter report, including the proportion of low abundance clusters and sequences that were removed; Supplementary File S4: the FROGS v3.2 ITSx summary report, including the proportion of low abundance clusters and sequences that were removed, in addition to the OTUs removed by sample; Supplementary File S5: the FROGS v3.2 taxonomic assignment summary report, including the taxonomy distribution across samples; Supplementary File S6: the complete list of the 722 ITS2 clusters generated using FROGS v3.2 and passing quality filters—tab "FROGS_buckhorn_ITS2_fungi". The second tab—FROGS_buckhorn_ITS2_LICHENS"—contains a list of the clusters representing lichen-forming fungi, with candidate species represented in only a single sample highlighted in orange; Supplementary File S7: Lichens represented in a polygon-constrained search of herbarium records from the San Rafael Swell region—ca. 4500 km$^2$—comprising 309 records documenting 86 taxa, spanning collections from the 1950's to the present CNALH; https://lichenportal.org/cnalh/ (accessed on 8 August 2022).

**Author Contributions:** Conceptualization, J.R.H., B.M.T., A.A.Y. and S.D.L.; methodology, J.R.H., B.M.T., A.A.Y., M.K. and S.D.L.; validation, S.D.L.; formal analysis, J.R.H., B.M.T., A.A.Y. and S.D.L.; investigation, J.R.H., B.M.T., A.A.Y. and S.D.L.; resources, S.D.L.; data curation, M.K. and S.D.L.; writing—original draft preparation, J.R.H., B.M.T. and A.A.Y.; writing—review and editing, J.R.H., B.M.T., A.A.Y., M.K. and S.D.L.; visualization, J.R.H., B.M.T., A.A.Y. and S.D.L.; supervision, S.D.L.; project administration, S.D.L.; funding acquisition, S.D.L. All authors have read and agreed to the published version of the manuscript.

**Funding:** This research was funded by the Department of Biology at Brigham Young University.

**Institutional Review Board Statement:** Not applicable.

**Data Availability Statement:** All short reads are available in NCBI's Short Read Archive under PRJNA866119.

**Acknowledgments:** We thank Mikel Baugh, Abigail Robison, Noppparat Anantaprayoon, and Larry St. Clair for help in the field and fruitful discussion that improved this study.

**Conflicts of Interest:** The authors declare no conflict of interest.

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
