# Peer review of "Characterizing Crustose Lichen Communities—DNA Metabarcoding Reveals More than Meets the Eye"

_diversity, doi:10.3390/d14090766_

Round 1

Reviewer 1 Report

The manuscript of Henrie and co-authors shows an elegant study about the use of DNA metabarcoding for deciphering complex (and often tiny) communities of crustose lichens in south-western USA. These types of lichens are usually neglected in most voucher-based studies of lichen diversity, so that the proposed method seems suitable for recovering an important proportion of the lichen diversity, especially in arid areas, where these lichens thrive. The sampling was well designed and the obtained results are promising, in the sense of using the same approach for the study of these communities in other parts of the world.

The manuscript is well written. I recommend this manuscript for publication. However, I would like the authors to address the following issues.

WRITING

Although I am not a native English speaker, I would suggest reviewing lines 27-28 in abstract “The interpreting these results…”. Probably it would be better to say “Interpreting these results…” or “The interpretation of these results…”.

I would also change to capital letter the “t” in “C. the location of the sample site…” (line 119, Figure 1 legend), and the “c” in “B. comparisons of candidate species…” (line 251, Figure 4 legend). Just for mantaining a coherent style in figure legends.

I am of the opinion of writing all scientific names (at any taxonomic rank) in italics, to facilitate communication (see Thines, M., Aoki, T., Crous, P.W. et al. Setting scientific names at all taxonomic ranks in italics facilitates their quick recognition in scientific papers. IMA Fungus 11, 25 (2020). https://doi.org/10.1186/s43008-020-00048-6). Thus, I would use this formatting when writing lichen family names.

INTRODUCTION

Even though this paper belongs to a special issue of Diversity journal about lichen, W would strongly recommend authors to explain in the Introduction section what a CRUSTOSE lichen is. This would facilitate communication for a wider audience (not only lichenologists). Authors may want to include a further figure (picture) showing a typical community of crustose lichens that they sampled (in case they made any photo in the field).

RESULTS

Lines 255-256. Here, authors say that “candidate species observed in a single SAMPLE are highlighted”, whereas in line 380 they say “in a single SITE”. I guess that “site” should be replaced by “SAMPLE”. Furthermore, I tried to deduce (without success) why, in supplementary file S6, there are some taxa highlighted in pale blue in column entitled “Bold-based taxon”. I suggest authors explain it in the corresponding supplementary file legend.

In Supplementary file S6 I found a “paulicaulina”. Perhaps they wanted to say Polycauliona?

GENERAL COMMENTS AND DISCUSSION

1) Lines 203-205. Authors indicate that they are aware of the potential caveats of identifying lichenized fungal species just by sequence similarity, which I totally agree. This leads me to think that the authors might have mentioned, in Discussion, the existence of another level of training related to the study they have conducted: the need to have an understanding of current tools for phylogeny reconstruction, and that this process could, or should, be tutored by lichenologist with at least some expertise in molecular phylogenetics. Furthermore, would the authors suggest employing some rapid and reproducible species delimitation method based on the ITS barcode to support (or help in) the "molecular inventory" of lichens? For example, ASAP, GMYC, etc? Or would it be detrimental (in the sense of including more noise)?

2) Authors mention that “ca. 63% of candidate species recovered at the Buckhorn Wash sites were present in the BOLD regional ITS databases” (lines 237-238). Was the professional lichenologist (SDL) aware of any species that he saw and collected in the field that later was not present in the ITS2 sequence dataset? I am in fact curious about Squamarina lentigera, which is a relatively “big” lichen, easy to spot in the field, that I would have expected to be in the sampling site. Wasn’t it there?

3) In my opinion, it would be expected that bulk samples would have a higher biomass of the more conspicuous species, especially those collected by technicians who do not have much experience in collecting lichens in the field. My question is: how do the authors assess the effect of a more or less balanced composition in the bulk samples (in terms of lichen biomass) on the results of Illumina sequencing and subsequent bioinformatic processing? Do they consider that the larger crustose lichens will be overrepresented (in terms of the number of reads), and consequently, the rarer species will perhaps disappear in the bioinformatic filtering processes? I am aware of the fact that finding a Usnea in the assembled dataset suggest that there will not be dramatic effects even if the samples are quite unbalanced.

4) Have the Authors examined if the lichens that correspond with the fraction of the regional lichen diversity that was not recovered with the metabarcoding approach have any trait in common? The size of the thalli, perhaps? Or do they think that problems with primer binding sites would be more problematic?

5) I found interesting differences between PROF1 and PROF2? Do Authors think it is a matter of the number of participants (just two persons including SDL in PROF1 vs three persons in PROF2)?? Or do they think it has more to do with the experience gained by SDL in his first visit to the sampling site, which probably allowed him to focus on more singular microhabitats hosting more diverse lichen assemblages in his second visit?

6) Just out of curiosity. Would it be possible to apply a similar methodology to the study of symbiont microalgae? And, if so, which primers would you suggest to use? And, as for the bioinformatics pipeline, would you also use FROGS?

Reviewer 2 Report

This paper is a good example of the use of metabarcoding in an environment where microlichens are a dominant component. It provides a comparison between bulk collections in a specific area by an expert and by technicians with minimum training which allows intraspecific diversity to be estimated within the selected site. This is then compared with a regional database of lichens to allow taxonomic diversity to be estimated. However its efficacy depends on the existence of a regional ITS database that can be used to convert the data into species that could be monitored in future, so its practical application is restricted to areas where there is already expert data available on the DNA of the lichen components in the region. The authors acknowledge the problems of this method including the lack of vouchers but point out that this method can lead to subsequent targeted sampling of specimens as well as to further analysis of the data from bulk sampling. I agree with this but point out that we are still very behind in accumulating regional lichen DNA data in remote areas, and that data from bulk sampling has to be backed up with representative vouchers in order to make conclusions about ongoing changes in community structure in response to environmental changes.
